# Saliency Driven Gaze Control for Autonomous Pedestrians

Category: Research

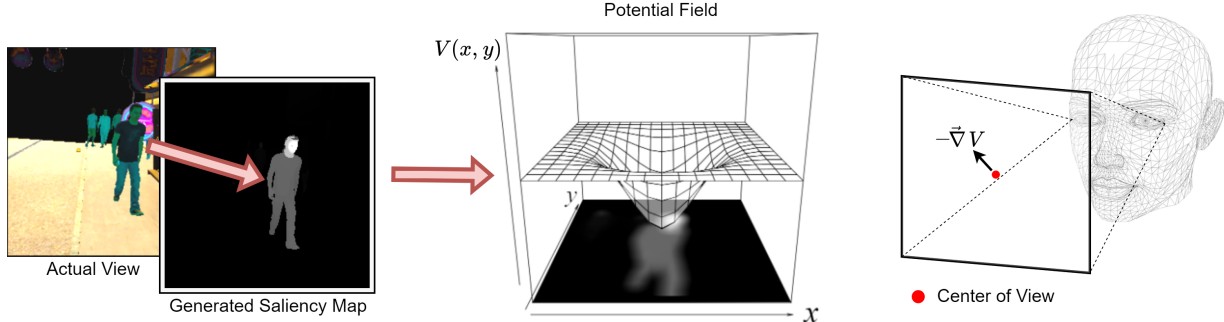

Figure 1: Components of the proposed Particle Gaze method. A saliency map is generated in real time from the current agent view (Left). The values from this saliency map are used to set the Z values of a spline surface, representing a potential field $V(x,y)$ (Middle). The gradient of this potential field $-\vec{\nabla}V$ is used to move the center of gaze towards a minimum in the field.

## ABSTRACT

How and why an agent looks at its environment can inform its navigation, behaviour and interaction with the environment. A human agent's visual-motor system is complex and requires both an understanding of visual stimulus as well as adaptive methods to control and aim its gaze in accordance with goal-driven behaviour or intent. Drawing from observations and techniques in psychology, computer vision and human physiology, we present techniques to procedurally generate various types of gaze movements (head movements, saccades, microsaccades, and smooth pursuits) driven entirely by visual input in the form of saliency maps which represent pre-attentive processing of visual stimuli in order to replicate human gaze behaviour. Each method is designed to be agnostic to attention and cognitive processing, able to cover the nuances for each type of gaze movement, and desired intentional or passive behaviours. In combination with parametric saliency map generation, they serve as a foundation for modelling completely visually driven, procedural gaze in simulated human agents.

**Index Terms:** Computing methodologies—Agent / discrete model; Computing methodologies—Procedural animation

## 1 INTRODUCTION

Modelling and simulating human gaze is a complicated endeavour. There are many approaches for estimating and approximating how a human agent may observe the world, most of which aim to replicate it to create believable appearing humans. These methods are effective and often take advantage of scene information from the simulation to calculate believable gaze patterns. However, if one's goal is to replicate gaze, not starting at the desired end result but from the basis of vision then it is important to try and follow a guideline of 'sensory honesty'; by generating gaze movements in an agent-encapsulated manner. In other words, without using information or data that a real human agent would not have. This work presents two methods for controlling gaze using only visual information as input. Specifically, saliency maps are used as input representing pre-attentive processing of the human psycho-visual system of visual stimulus.

There are many factors and complications to consider when presenting a model of human gaze. As such, we construct a framework for authoring a variety of gaze behaviours. We use two modes of controlling gaze with saliency maps as input to develop a novel method which can cover a wide range of human gaze behaviours.

Thus far, gaze behaviours in crowd simulations have been largely absent or homogeneous. A robust gaze model requires more than just saliency maps. Once a saliency map has been generated, how do we determine which targets to look at, the order in which they should be gazed at, and the duration of the fixation, in such a way that the resulting behaviour and animation is robust and convincing, and also in a way that can be adjusted depending on the situation? This work proposes two models that provide an authorable framework for designing a diverse set of gaze behaviours that can be adjusted on a per-agent basis, promoting heterogeneity in crowd simulation gaze behaviours.

Our contributions are as follows. First we present a particle gaze model that uses a potential gradient field to drive gaze towards salient regions of the saliency map. Second, we present a second probabilistic saccade model that chooses targets from the saliency map probabilistically and executes quick saccades, and is capable of making microsaccades by then choosing fixation points based on saliency in the localized region of the target. Third, we evaluate our particle model against pyStar-FC, a notable multi-saccade generator and demonstrate that our model can be tuned to a high degree of similarity with other models. Finally we compare our two models against each other qualitatively.

## 2 HUMAN GAZE

Human gaze is a complex topic, weaving between physiology and psychology. Consequently, a model of gaze that neglects either aspect will be woefully incomplete. One of the core points of this work is to accurately replicate the suite of human gaze movements along with their subtle nuances, which involves understanding the mechanics, limitations and strategies of how people look at things. This is discussed in more detail in [4].

### 2.1 Gaze Movements

Human eye movements have been the subject of intense study for many decades. Over this time eye movements have been classified into 7 categories. The standard set of eye movements are (from [36]):

- **Saccade**: voluntary jump-like movements that move the retina from one point in the visual field to another;

- **Microsaccades**: small, jerk-like, eye movements, similar to miniature versions of voluntary saccades, with amplitudes from 2 to 120 arcminutes;

- **Vestibular-Ocular Reflex**: these stabilize the visual image on the retina by causing compensatory changes in eye position as the head moves;

- **Optokinetic Nystagmus**: this stabilizes gaze during sustained, low frequency image rotations at constant velocity;

- **Smooth Pursuit**: these are voluntary eye movements that track moving stimuli;

- **Vergence**: these are coordinated movements of both eyes, converging for objects moving towards and diverging for objects moving away from the eyes;

- **Torsion**: coordinated rotation of the eyes around the optical axis, dependent on head tilt and eye elevation.

There are quite a few different types of eye movements, each with its complexities and implications on human gaze. For the sake of this work, we focus on how to model saccades, microsaccades and smooth pursuits. Vestibular-ocular reflex is responsible for stabilizing the visual image as the head moves. In addition to eye movements, head movements are also a crucial part of gaze. We label the set of eye and head movements as *gaze movements*. Due to the slower nature of head movements, these tend to be less categorized. As a general rule, humans tend to align their heads with what they are looking at. According to [25], this is because a discrepancy between the head and eye directions causes interference in visual processing, as well as a degradation in accuracy for localizing attentional focus and hand-eye coordination. Head movements are less erratic than saccades or otherwise would cause strain on the human neck. One study [5] found that head movement duration can range between 200-800 ms when a series of saccades make up a gaze shift, with larger head rotation speeds for larger gaze shifts. In contrast, a single saccade-fixation action requires about 200 ms. In [6], a saccade took just under 200 ms while a head movement took just under 450 ms to complete in a single trial, suggesting that head movements are generally slower than saccades. Most natural gaze shifts utilize a combination of saccades and head movements, with head rotations typically following the eyes with a delay of 20-50ms [32]. A model which aims to emulate human gaze should be able to parameterize and replicate these types of gaze movements, or at least a sufficient subset of them. The large problem with generalizing a control structure however is that human gaze behaviour tends to be very diverse and idiosyncratic [32]. The selection of gaze targets are drawn from attention and deliberate intent, which then informs the gaze. For example, a slow-moving object of interest in view will elicit a smooth pursuit. However, if this target is moving too fast smooth pursuit is no longer possible and the human visual system will resort to "catch-up" saccades to keep track of the object. A model of gaze should be able to generate a range of plausible eye movements given knowledge or a map of how the given agent is attending to their world.

## 2.2 Memory and Inhibition of Return (IOR)

Inhibition of Return (IOR) is described by [15] as a delayed response to stimuli in a peripheral location which was previously attended to or looked at. Originally found in [30], and followed up by and defined in [31], IOR's function appears to be orienting gaze towards novel locations which facilitates foraging and other search behaviours. This is fairly intuitive, e.g. if you were searching your office for a specific item it would make sense to avoid searching where you have already looked. Alternatively, if you were just trying to gather information about your environment, the same mechanism aids in information gathering. IOR typically appears in the literature when the stimulus event is not task-relevant or there is no task given to the observer [15]. When test subjects were tasked with making saccadic movements which seemed most comfortable after viewing a brief stimulus they most often would look away from the location of the stimulus. [15, 16] found that across multiple studies it appeared that IOR is often encoded in environmental coordinates rather than retinal coordinates. This effect appears in the early IOR literature [30, 31]. Further studies have also shown that in some instances IOR appears to be encoded on an object basis [1, 8, 34, 35]. Both environmental location and object attachment as IOR encodings have strong experimental evidence to support them. The question becomes, in what cases do either occur? [15] suggests that this change in encoding occurs depending on contextual factors such as whether the observer is moving, objects in the view are moving and what the intent or task of the observer is. In early studies, IOR appeared as related to a reluctance of motor response to focus on particular locations, not inhibiting or suppressing attention. However, studies have found IOR to occur in spatial tasks as well, not just stimulus-response. These findings have shifted the general consensus that IOR does indeed occur on an attentional level as well as oculomotor response. The reasoning again appears to be contextual. For example, the type of stimuli, as well as the difficulty in discriminating stimuli within an observer's view affects the introduction of IOR on the attentional level. The presence of IOR on attention is further supported by findings that IOR also appears in auditory [23, 24, 33] and tactile [34] modes. Results are consistent in demonstrating that IOR's effect is to inhibit responses typically associated with stimuli. Narrowing down how IOR mechanisms will function is a difficult task affected by many factors. Studies have found generally that IOR typically takes between 100ms and 200ms of cued saccade fixation to kick in which aligns with the time between saccades which typically has latencies of 200-250 ms [7]. The effects can last several seconds, however, this can easily be affected by changes in the scene or task of the observer.

The multitude of open questions as well as contextual changes makes it difficult to define an inhibition of return mechanism which can be used as a part of a gaze control system. Factors like environment, agent factors, intent, and task all need to be taken into account to decide for example what kind of encoding to use. That's not to mention open questions on specific mechanisms within IOR. There are many valid ways to implement IOR, in this paper we try to focus on one subset of factors and contexts and suggest an IOR mechanism contingent on that based on the previously mentioned literature. It is the hope as well that implementing gaze control systems based on attention and vision literature opens up possibilities to explore many of the open challenges yet unexplained.

## 3 RELATED WORKS

To create a full pipeline modelling the gaze of an agent requires first defining *what* to look at and then *how* to look at it. Our work uses saliency to capture a visual representation of what/where an agent may look at. Deciding how to use saliency information to generate fixations or eye movements is an area of ongoing study. Though the field is mostly saturated with models for predicting fixations within 2D images, we draw inspiration from other works in this area and apply concepts to our problems for 3D characters in a dynamic simulation.

## 3.1 Saliency

Saliency models attempt to represent what is important within a field of view, typically concerning human visual processing. The most common form of representing this is in the form of a *saliency map*, a 2D image which describes which regions within a field of view are "salient". In this sense, saliency is usually interpreted as

the probability that a human observer will look in a particular location. Rule-based models such as [11]'s originative work construct saliency maps based on things like colours, contrast, location etc. More recent deep learning models like [12] and [20] aim to emulate human saliency specifically by training off of human scan-path data on sets of images. However, the main issue with these models are inherent biases within datasets, and in the case of human simulation, they are prohibitively slow to use in real-time simulation. Virtual saliency models are aimed at implementing saliency specifically for the purposes of human or agent simulation. For example, it is possible to construct a model of saliency from a simulation scene database and assign scores to objects within an agent's view [26]. This is a simple and effective approach however is limited in its uses outside of simulating visually believable gaze animations. To go beyond these limitations it is possible to use a rule-based model for generating saliency maps (parametric saliency maps) in real-time during a simulation using information from embedded in the scene graph and localized per observer [17]. The advantage of this approach is bringing saliency maps to real time-simulation, which means vision-based approaches to gaze control and scene understanding are possible. In follow-up work, the parameters of the parametric saliency map were learned by minimizing the output difference from state-of-the-art deep saliency models on a virtual dataset [18]. Work has shown that visual attention is guided by features depending on the task, and that pre-attentive features like colour, luminance, motion, orientation, depth, and size are all key elements of visual attention [39]. All of these can be compactly encoded into parametric saliency maps, which is why it is an efficient representation of pre-attentive processing for attentive tasks like fixations. Similarly, work has shown that bottom-up features (stimuli) guide attention under natural conditions, for example, simple undirected gaze with no intent or goal [27].

## 3.2 Fixation Prediction

Many findings summarized in [40] conclude that saccadic selection avoids areas of little or less structure within an image. When compared with random fixation point selection on datasets of images, regions chosen by actual fixation locations have consistently higher signal variance than random selection. [41] found that the mean-variance ratio of random vs. real fixations $\sigma_{eye}^2 / \sigma_{rand}^2$ to be around 1.35. Active fixation prediction from [37] aims to generate a temporal series of fixation locations in an image which can be used to construct scan paths. They accomplish this through a tiered saliency approach, blending a coarse feature map on the periphery with a high detail saliency map located at the point of fixation. Combining this with a temporal inhibition of return mechanism (IOR) they are able to generate very plausible scan paths. Notable takeaways from this approach are the importance of selective suppression of attention or saliency in the periphery, combined with some mode of memory to implement inhibition of return which from [16], says is consistently found in studies of fixations and saccadic eye movements. The recent Deepgaze III model from [19] trained a deep neural network to predict and generate scan paths and fixations from fixation density maps (i.e. saliency) for free-viewing of natural images. The model generally outperformed other similar models (such as the previously mentioned STAR-FC) in various statistical measures on state-of-the-art datasets. The model is particularly interesting due to the modular architecture allowing them to conduct ablation studies to quantify the effects and relevance of input data. It was found that scene content has much higher importance on fixation prediction than previous scan path history. As noted by Tstotsos, J., one key limitation is the static nature of images and how shifting of gaze does not affect the image. Key challenges we address are how to implement inhibition of return given dynamic environment, agent position and agent gaze. As well as how to select fixation points.

The problem generally with all these approaches is the focus on free viewing of static images. That is useful for trying to predict how

someone may look at an image, however, as noted above, humans do not see in 2D static images. Human visual systems contend with stimuli changes from dynamic environments as well as egocentric effects when gaze movement occurs (i.e., changing where you look completely changes the information available to your vision). The pursuit of fixation prediction in active-vision applications; such as simulation or robotics, must contend with temporally changing environments, changes in agent position, changes in agent gaze orientation, and spatial-temporal memory.

## 3.3 Gaze Control

One of the closest implementations of our approach is [29], where the Itti-Koch-Niebur (IKN) model [11] is used to generate saliency maps from the perspective of a virtual agent. This was used to determine which objects within view would be 'salient' and queued them as targets in the scene database. They also implemented a form of memory where agents would keep track of scene objects that they have observed. The spirit of their work was 'sensory honesty', in trying to use as little simulation knowledge as possible. In the same vein, our work also shares this same goal but attempts to take it further by including no information about the transforms of objects in the scene database in gaze, having it entirely driven by visual stimulus. The most significant limitation of the authors was the lack of a top-down attention component. This is addressed by our inclusion of parametric saliency maps from [17]. The benefit of using saliency maps is that the processing time is limited only by the cost of the attention model and the rendering pipeline. Another limitation is simply that humans don't have a scene database to draw information from. Approaching the problem of gaze and attention from a visual stimulus-driven standpoint opens the door for more grounded modelling of virtual humans. More complex totalistic models for automating gaze behaviour have been worked on for over two decades, in the form of cognitive models of attention and intent which form a high-level controller [3, 14, 22]. These models rather interestingly attempt to join ideas of task relevance and action to inform gaze movements. This is an often-overlooked factor despite environmental conditions impacting visual understanding of the environment which also impact general locomotion and movements, such as the increase in foot clearance on steps in different lighting levels [10]. Our proposed approach poses a simpler parametric framework for authoring and generating gaze behaviours in a way which compartmentalizes attention and intent away from control. Our work fits in as a link between the vision-based approaches, like [29], and high-level control structures, such as [3, 14, 22]. Other pseudo-saliency driven gaze approaches do not use visual stimulus as input control for gaze by explicitly targeting transforms of objects within the scene [2, 26]. These approaches create reasonably believable procedural gaze animations however are limited to scale as a scene becomes increasingly complex the computational costs of such gaze models too will increase.

Gaze behaviour modelling is not only important in real-time applications but for a variety of purposes. For example, gaze behaviour can be inferred from motion capture data and automatically integrated into animation as done in [28].

A recent paper proposed a real-time method for driving gaze behaviour using a multi-layered saliency approach similar to ours [9], but it does not take into account 3D information from the scene such as velocity of agents, and the customization maps seem to be created for an entire viewpoint rather than for individual objects, so new customization maps would have to be made for each viewing direction of an object whereas our method allows semantic masking that is attached to the object and works for any viewpoint. Additionally, the use of a ML saliency model limits the customizability of the saliency maps, whereas our method uses PSM [17] which provides great flexibility and authorability.

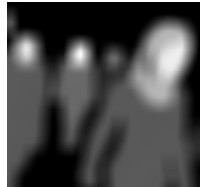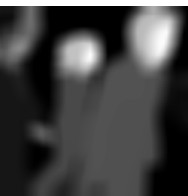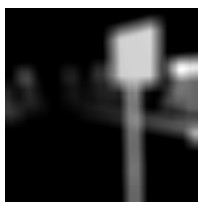

Figure 2: Examples of generated saliency maps from the perspective of an agent walking through a simulated urban crowd, using PSM weights specified in [18].

## 4 METHODS

Each method presented takes as input a square, grayscale image representing the saliency of an agent's view at that time, and then outputs a new orientation and the speed at which to interpolate to it from the current orientation. Once the new target orientation is reached, the process is repeated. Each method is designed to be simple, yet capable of plausibly generating different types of gaze movements. At the same time, they are agnostic to top-down attention which is instead encoded in saliency maps. Through the combination of saliency and the control parameters for the gaze-control methods, a wide range of intentional and passive gaze behaviours can be modelled.

We use the Predictive Avoidance Model (PAM) [13] for agent navigation, which senses obstacles and neighbours within some field of view and produces piece-wise predicted repulsive forces to avoid them. Our gaze behaviour models change the center of the field of view, which affects the neighbours and obstacles avoided. Further selecting avoidance targets based on saliency is planned future work.

### 4.1 Saliency Map Generation

We utilize the parametric saliency maps (PSMs) method from [18]. This allows for saliency maps to be easily generated in real-time for virtual agents. PSM is a compact way of encoding pre-attentive and top-down factors. Parameters can be easily adjusted to suit different attentive loads. The saliency of an object from the perspective of an observer is computed from the combination of weighted parameters,

$$S = W \cdot (w_d S_d + w_F S_F + w_v S_v + w_R S_R + w_I S_I) \cdot (W_M S_M) \cdot (W_A S_A)$$
(1)

The factors determining object saliency include depth $S_d$, orientation saliency $S_F$, normalized speed $S_v$, normalized angular speed $S_R$, interestingness value $S_I$, sub-texture masking $S_M$, and visual attention weighting $S_A$, each parameterized by $w_d$, $w_F$, $w_v$, $w_R$, $w_I$, $W_M$, and $W_A$ respectively, subject to their respective constraints. For more details about these terms, see [17].

The values of weights are set by the observer. The parameter values come from the objects in the scene. For example, the interestingness factor $S_I$ is an intrinsic value from an object/character. It is an effective way to generate saliency maps in a simulation and change the attentive factors as needed, either globally through factor values, or on a per-agent basis through the factor weights. A Gaussian blur is applied afterwards to smooth out hard edges.

### 4.2 Particle Model

We now introduce the particle model for saliency-driven gaze control. This model treats the center of gaze as a particle which is acted on by driving forces. By imagining the center of gaze as a particle in a potential field we can use equations of motion to describe how it moves. The potential field comes from the saliency of what the agent is seeing.

### 4.2.1 Particle Update

The point which lies in the center of view (from a virtual camera) can be imagined as a point on the 3D viewing sphere around an agent. Moving this point around the sphere is equivalent to changing the direction in which an agent is looking. Treating this point as a particle, gaze "forces" can be applied to it which change the direction of gaze.

For a given discrete time step $t$, an agents gaze state can be described by $\mathbf{G}_t = (\theta, \phi)$, which represents a point in spherical space for a fixed radius, where $(0,0)$ is the natural or forward-facing orientation. For a saliency map $S_t$; which represents the current view's saliency map, a potential field is defined as $V(\mathbf{G})$. By interpreting points of high saliency as potential wells in $V$, following the gradient will drive the gaze-particle into highly salient regions. We can formulate the motion of the particle as,

$$\ddot{\mathbf{G}} = -\vec{\nabla}V - k_d\dot{\mathbf{G}}$$
(2)

Where $-\vec{\nabla}V$ is the force applied by the potential to the gaze particle based on what the agent is currently seeing in the current saliency $S_t$. The term $-k_d\dot{\mathbf{G}}_t$ represents damping with coefficient $k_d$. The algorithm to update the position of the particle for step size $\lambda$ is given by,

$$\ddot{\mathbf{G}}_t = -\vec{\nabla}V(\mathbf{G}_t) - k_d\dot{\mathbf{G}}_t$$
$$\dot{\mathbf{G}}_{t+1} = \dot{\mathbf{G}}_t + \lambda \cdot \ddot{\mathbf{G}}_t$$
(3)
$$\mathbf{G}_{t+1} = \mathbf{G}_t + \lambda \cdot \dot{\mathbf{G}}_{t+1}$$

Additionally, we can include a noise term $A \cdot z_t$; where $z_t \in [-1,1]^2$ with amplitude $A$, in the final position update which gives added flexibility to model more complex gaze movements. The final update is then,

$$\mathbf{G}_{t+1} = \mathbf{G}_t + \lambda \cdot \dot{\mathbf{G}}_{t+1} + A \cdot \mathbf{z}_t$$
(4)

An important consideration then is how to construct the potential fields from the saliency maps. Looking at examples in Fig. 2, one problem is that in most saliency maps there are large regions of little or constant saliency. This presents a problem because there would be no gradient in these regions. Another thing to consider is that highly salient stimuli should draw gaze towards it regardless of where it is in the visual field. Of course, the method should be computationally efficient in order to scale for large groups of agents. Calculating the potential field from an $n \times n$ image could be very costly, especially scaled to scenarios with many agents. The solution we chose is to use a parametric surface to model the potential by sampling from the saliency map. Cubic B-splines have useful properties which make them very effective and efficient for this task. Assuming an appropriately chosen number of control points, a cubic B-spline surface will have a non-zero gradient in almost all regions of the space, as well as being very fast to compute. Sampling from the saliency map, the heights of control points on a spline surface can be set giving a reasonable approximation of a potential field. $\vec{\nabla}V(\mathbf{G})$ is then the gradient of the surface with respect to the *x-y* plane.

The max-pool and average-pool algorithms are commonly used in computer vision to downscale images into a lower resolution space. We use average-pool to pool values from a saliency map into the control points of a spline surface. For a lattice of $m \times m$ control points, the saliency map is divided into $m \times m$ windows. The heights of control points are set as negative results of pooling each window, with the maximum depth being -1. At each time-step $t$ the control points of the surface are set, giving the potential field. Fig. 3 shows a simple example for an $m = 7$ surface, while Fig 4 shows the projection of the environment onto the agent's view. Since the gaze-particle is always at the center of the visual field, the gradient is

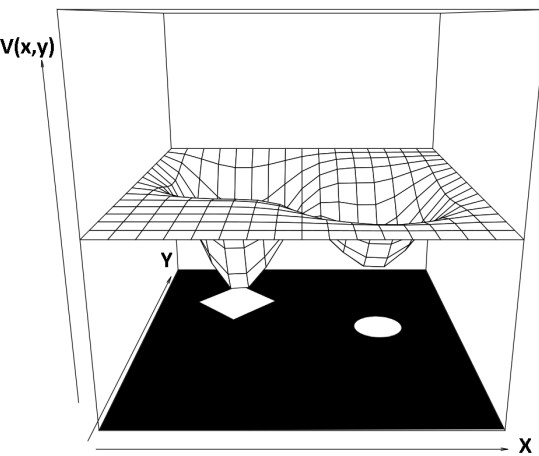

Figure 3: Spline surface representing the potential field. Values from the saliency map are pooled into control points corresponding to quadrants.

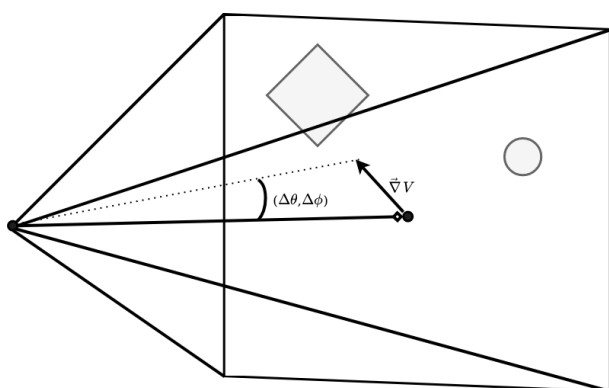

Figure 4: Projection of environment onto the agent's view. $\vec{\nabla}V$ is the gradient at the center of the potential field. For small values, $\vec{\nabla}V \simeq (\Delta\theta, \Delta\phi)$, where $(\Delta\theta, \Delta\phi)$ are the updates to the current camera orientation. This moves the view until the center is in a local minimum (Typically corresponding to the center of an object of interest)

always sampled at the center of the potential field as well. Following Eq. 3, the gaze-particles position on the viewing sphere is updated, changing the point of view.

We chose to use a 3D spline surface over other traditional 2D methods because it provides an intuitive means for adjusting the behaviour. For example, instead of directly defining interpolation behaviour, one can simply adjust the 3D spline parameters, and the interpolation behaviour will automatically adjust. By adjusting the parameters of the spline surface, we can get a continuous gradient without the issue of gradient deadzones.

### 4.2.2   Control

The primary parameters for control are the step size $\lambda$ and damping coefficient $k_d$. A large step size will cause the view to move quickly through the visual field however will struggle to stay on target. A small step size will have excellent tracking of targets once fixated but will struggle to move to new targets. For this, we propose a two-state system for varying the behaviour of the particle's movement. In the *search* state, the step size is set to $\lambda_{search}$. The gaze is free to move around and will be drawn in by salient regions in the view.

---

**Algorithm 1** Particle Gaze Model
> $STATE \leftarrow search$
> $G \leftarrow (0.5, 0.5)$          ▷ Center of viewport
> $\dot{G} \leftarrow (0, 0)$
> **while** *true* **do**
>      $V \leftarrow \text{SetPotential}(S_t)$
>      $\ddot{G} = -\vec{\nabla}V(G) - k_d \cdot \dot{G}$
>      **if** $STATE == search$ **then**
>          $\lambda \leftarrow \lambda_{search}$
>          **if** *FixationDetected()* **then**
>              $STATE \leftarrow fixation$
>          **end if**
>      **else if** $STATE == fixation$ **then**
>          $\lambda \leftarrow \lambda_{fixation}$
>          **if** $fixationtime > \tau_{fixation}$ **then**
>              $STATE \leftarrow search$
>          **end if**
>      **end if**
>      $\dot{G} \leftarrow \dot{G} + \lambda \cdot \ddot{G}$
>      $G \leftarrow G + \lambda \cdot \dot{G}$
> **end while**

---

As the particle moves into a potential well, the gradient will get smaller. At this point, there needs to be some definition for detecting a fixation, which should work regardless of motion either egocentric or by the target object. We define a simple rule which measures the average gradient of the potential within some temporal window. If the average gradient has dropped below a threshold then a fixation has occurred and the state is changed. In this state, step size is set to $\lambda_{fixation}$, and this state lasts for $\tau_{fixation}$ seconds.

After the fixation time, the saliency of the target would still affect the potential field, thus it is important to implement an inhibition of return mechanism to prevent getting stuck on one target. For the parametric saliency maps we utilize, the saliency of targets under the particle can be decayed. This simple rule allows the particle to move on to new targets naturally, encoding object-based IOR. A good default value is a decay time of 1-2 seconds for general searching/foraging gaze behaviour, however, it must be noted that for accurately replicating specific gaze behaviours this value would likely need to be different depending on need. There is also some added complexity to consider in how exactly saliency returns after it has been decayed, however for the scope of this paper we do not discuss what/how this might be done as for general use, targets will be well out of view before IOR would wear off. If one imagines walking down a busy street people, cars, signs etc. will constantly be coming in and out of view, so we feel this rule is sufficient.

There are properties of the particle model which lend themselves well to controlling head movements, as well as smooth-pursuit eye-movements. First, is the naturally smooth motion which arises towards targets of high interest. Second, for a small number of control points; recommended 7 for a degree 3 spline surface, this method has the natural tendency to align with general areas of high interest at low resolution. This often means looking at the "center of mass" of areas with high saliency targets as opposed to specific individual elements if there are many within view. If there are sparse, spaced-out objects of interest the gaze will instead align with the individual elements. Both these behaviours arise without explicit programming. Setting the points in the control surface to a higher resolution will yield more spacial acuity, and thus the gaze will fall on more narrow targets. Changing the step size $\lambda$ will determine how fast the gaze will move towards targets, as well as how strongly those targets will be tracked. Smooth pursuit eye-movements can be elicited by having a high resolution in the control surface; recommended 11 for a degree 3 spline surface, and a larger $\lambda_{fixation}$ value. It is difficult to recommend any particular value for

$\lambda_{fixation}$ because this will be scaled with how the spline surface is defined, steep peaks are as well as how fast objects move across the field of view which is limited by the frame rate of a given simulation. The length of smooth pursuits is something contextual. For a typical "search" behaviour, the length of fixations $\tau_{fixation}$ should average $150 - 300$ ms. For saccadic movements, a larger $\lambda_{search}$ value will give faster rapid target acquisition. To emulate micro-saccades we can peturb the final position using a noise term $A \cdot z_t$, where the amplitude corresponds less than $0.1°$ of visual angle. This will depend on camera projection parameters, but a small angle approximation $A \simeq 0.1°$ is acceptable. Additionally to improve accuracy and avoid oscillations, multiple steps can be taken per simulation time step. In the scope of this work we do not describe how to switch between saccades and smooth pursuits. This is largely because smooth pursuits are typically *intentional* actions and need to be specified by the author of the behaviour.

### 4.3 Probabilistic Model

#### 4.3.1 Target Selection

In this section we introduce another method for saliency driven gaze control, based largely on prior works in fixation prediction for static images. A saliency map can be thought of as a probability distribution for likely gaze targets. With this interpretation, fixation targets can be sampled from this distribution. For a probability distribution $S_t$, a random point $\mathbf{x} \sim S_t$ is drawn. Based on the projection parameters of the virtual camera, this point in the viewing image can be converted to an orientation. The agents view can then be rotated accordingly to match this orientation.

#### 4.3.2 Control

---

**Algorithm 2** Probabilistic Gaze Model

---
**Def:** *LookAt*(*point*, *time*)
$STATE \leftarrow search$
$G \leftarrow (0.5, 0.5)$                    ▷ Center of viewport
**while** *true* **do**
   **if** $STATE == search$ **then**
      $\mathbf{x} \leftarrow SamplePoint(S_t)$
      $LookAt(\mathbf{x}, \Delta t_{saccade})$
      $STATE \leftarrow fixation$        ▷ Wait until reached target
   **else if** $STATE == fixation$ **then**
      $S_W \leftarrow S_t.window(R_{focus})$
      $\mathbf{x} \leftarrow SamplePoint(S_W)$
      **if** $fixationTime > \tau_{fixation}$ **then**
         $STATE \leftarrow search$
      **else**
         $LookAt(\mathbf{x}, \Delta t_{\mu saccade})$
         $Wait(\tau_{\mu fixation})$     ▷ Hold for length of $\mu$-fixation
      **end if**
   **end if**
**end while**

---

Given a point $\mathbf{x}$ $S_t$ in viewport coordinates, a line can be drawn from the camera center through this point in world space. This vector represents an orientation $G'$. The current camera orientation $G$ can then be interpolated to this new orientation over a desired time. The speed of the rotation is then determined by the interpolation time.

Divide control into two primary states: search and fixation. In the search state, a point is sampled from the entire field of view. The view is then oriented to this target over $\Delta t_{saccade}$. The angular speed of the saccade is the amplitude (angular) divided by $\Delta t_{saccade}$. Once this target is picked the state transitions to fixation control. Over a total time $\tau_{fixation}$ saliency outside a small foveated region of radius $R_{focus}$ is suppressed. Within this fixation, new points are drawn from the foveated region of interest as targets for micro-fixations.

The point is then interpolated to over $\Delta t_{\mu saccade}$. This point is looked at for time $\tau_{\mu fixation}$, at which point a new target is selected. This repeats over the entire fixation length. Once the fixation has concluded, the state returns to search. Each parameter can be set statically or dynamically depending on desired behaviours.

This method of control is designed to allow modeling of target point selection saccade and micro-saccade eye-movements. Depending on the level of detail desired, keeping $\Delta t_{saccade}$ and $\Delta t_{\mu saccade}$ constant will achieve linear eye velocities expected for angular distances less than $20°$, which typically reach up to $300°/s$. However, for most applications it suffices to have a very small or zero travel time (i.e. instantaneous). Changing the $\tau_{fixation}$ parameter will affect how much searching is done in the visual field. Veering from typical reported values of around $100 - 200ms$ will result in either rapid eye-darting for smaller values, or more focused eye-movements in the case of larger values. Tightening or increasing the size of the focus region $R_{focus}$ will either restrict the space of micro-saccade movements (thus decreasing their amplitude) or allow for more outside stimuli to draw micro-saccades respectively. Depending on the desired behaviour either can be appropriate. For example, a character reading a book would have very infrequent saccades (large or infinite $\tau_{fixation}$), frequent micro-saccades (small $\tau_{\mu fixation}$, and a small radius of focus $R_{focus}$. Similarly to the particle method, we implement inhibition of return as a decay in object saliency.

## 5 RESULTS AND EVALUATION

Here we present evaluations of our models. First, it should be noted that the PSM saliency maps our models are predicated on have been previously evaluated against SALICON, a state-of-the-art machine learning saliency, with high correspondence [18].

We compare our particle model fixations against pyStar-FC [38], a notable multi-saccade generator. The pyStar-FC model generates saccades for static images, so we constructed scenarios in our virtual environment where neither the viewing agent nor pedestrian agents are moving in order to create static images for comparison. The gaze movement of the viewing agent can then be projected onto this static image to show the scanpath of the agent using our model. Then we compare this scanpath to the output of pyStar-FC on the same RGB image. Scanpaths are generated from pyStar-FC for RGB images by internally computing a saliency map of the image, and then outputting the scanpath on the original RGB image, similar to our approach. We used mostly default parameters for pyStar-FC, using Deepgaze II with ICF as the saliency model [21]. The input viewing size was modified to match the field of view of our agents. Changing the IOR (inhibition of return) decay rate parameter in pyStar-FC did not produce significantly different results, so it was left at default.

The results emphasize the authorability of our method. By adjusting our model parameters, our particle model can be tuned to match pyStar-FC's output or any other model. Ten pairs of images were compared, five of which are shown in Fig. 5. It is worth noting that our use case was not the intended purpose of either Deepgaze II nor pyStar-FC, so there may be biases in their output on our virtual images. The tendency of pyStar-FC to fixate on the neon signage is likely a result of bias in the datasets used to create these models, which probably used well-lit, non-virtual environments. Thus the output given by pyStar-FC may not be representative of what humans would look at while navigating in this environment. Regardless, our aim in this comparison is simply to illustrate the authorability of our model and show that by adjusting the parameters of our particle model, we can match the output of pyStar-FC or any other model with a high degree of similarity. Our model maintains authorability and control while being able to match other models, whereas pyStar-FC's parameters are less flexible and intuitive, and changes to the underlying ML saliency algorithm, Deepgaze II or SALICON, would require retraining.

Table 1: K nearest neighbour similarity scores for five trials comparing our method's fixation points with pyStar-FC's, where k=2.

| Trial | KNN Similarity |
|-------|----------------|
| 1     | 0.965          |
| 2     | 0.976          |
| 3     | 0.989          |
| 4     | 0.989          |
| 5     | 0.988          |

The model parameters and weights of both the saliency model PSM and the gaze controller were tuned by hand to match the output of pyStar-FC. For example, we increased PSM interestingness of objects where fixations were generated by pyStar-FC and decreased interestingness of objects where no fixations were generated by pyStar-FC. We also adjusted PSM layer weights. For example in images where pyStar-FC mostly focused on static objects, the weights of layers preferring high-velocity or rotational objects were lowered. For images where pyStar-FC preferred closer or further objects, the weights for the depth and visual attention layers were adjusted accordingly, etc. Additionally, we tuned the parameters of the gaze model, for example choosing appropriate number of points in the potential field to ensure continuity in the gradient for the chosen targets. It would also be possible to tune the model automatically using an optimization framework such as CMA-ES, similar to [18]. Particle swarm optimization technique could also be used. This could address some shortcomings in our model discussed in later sections.

As Star-FC is a static fixation generator however, certain gaze parameters such as decay rate or fixation duration have no influence on the result for this comparison, whereas saliency map parameters have large influence. Gaze parameters would have more influence in dynamic simulation comparison with head and eye movement.

We compared fixations from our particle model to pyStar-FC fixations using k-nearest neighbour similarity for the same five trials. The resulting knn similarity scores were all over 0.95 indicating a high degree of similarity. The results are summarized in Table 1. Thus we show that we are able to match other models with a high degree of similarity. Matching it to real human gaze data is important planned future work. However it should be emphasized that our goal is not to match human gaze data but to present a flexible and customizable system for authoring gaze behaviour in virtual agents, which we have shown.

We can make some comparisons between both models. Figure 6 shows target selection for both models. The particle model drives gaze in the direction of the potential field gradient. The probabilistic model identifies potential gaze targets highlighted with red circles, and chooses one probabilistically based on saliency at that location. Once the probabilistic model chooses a target, gaze is snapped to that location–similar to human saccades. Additionally, the ability to perform microsaccades is one of the defining features of the probabilistic model. Reducing the field of view once a target is selected produces a zoomed result from which microssaccade targets can then be selected. Figure 7 shows this zoomed effect in comparison with the particle model.

Our models also account for saliency decay. While an agent fixates on something in the scene, we perform a raycast in the fixation direction, which when it hits the target triggers the saliency for that object and that viewer to decay, and this continues over time while that target is fixated on. An example is shown in Figure 9, where two agents are viewing the same man with different saliencies due to saliency decay. Decay rates for both models differ in these examples, however can easily be parameterized to produce different gaze behaviours–such as nervous eye movements versus watchful gaze. While the decay rate here for the probabilistic model is fast in order to encourage quick saccades, the particle model was set to a slower decay rate. More research is needed to determine an optimal decay rate and this is important planned future work, and we hypothesize that these values relate to context and stylization of the behaviour. An in-depth statistical evaluation of our gaze models is planned future work.

We also note that our method provides for multi-agent saliency evaluation as shown in Figure 8. This affords complex scenes with a multiplicity of independent gaze controllers automatically driven by diverse scenes. That is, crowds respond naturally to the makeup of a scene from signage to fellow pedestrians.

## 6 DISCUSSION

The strength of this approach is that the user does not need to explicitly define gaze patterns, but instead only needs to define an agent's visual task or intent. One of the main principles of this work is creating control which adheres to the idea of sensory honesty. Prior works in the area of simulated gaze control have been able to create reasonably believable gaze movements for characters utilizing information from the simulation itself such as the scene database to locate gaze targets and track their position. The hope is that we can start to think of autonomous virtual humans and how they actively view their environment in terms such as their intentions, goals and knowledge. We could describe what they are attending to and what their visual task is without having to write explicit patterns for how they should then generate gaze movements. Perhaps the most obvious addition to our work is definition of high level control for generating saliency maps and appropriately selecting the correct control parameters. SDGC only provides only one part of a full solution for generating plausible gaze movements. Ultimately, this requires thinking about how saliency (attention) should be defined and the interplay with an agent's intent. It fundamentally changes how we view and approach gaze of virtual agents from asking, "*what is this character looking at?*" to instead asking, "*what is this character interested in, and what are they trying to do?*". In practical terms this is deciding how to define saliency, and deciding what kinds of gaze-movements to use. Of course, an obvious criticism is that due to this, no general solution is offered which covers all or a large number of gaze behaviours. Using an optimization framework may be able to mitigate this issue by finding parameters that work well across a range of use cases, or to find different sets of parameters that work well for specific cases and potentially could be set dynamically during runtime depending on the viewer's situation. However, even in light of this our framework does expand the capabilities of similar works like [29] by including top down pre-attentive component in the form of parametric saliency maps from [17] which allows encoding things like novelty or task relevance directly into saliency. High level controllers for automating attending behaviour such as the extensive work from [3, 14, 22] combined with our Saliency-Driven Gaze Control (SDGC) approach to create a totalistic saliency driven model which takes into account agent action and intent, and subsequently delegate saliency generation and SDGC methods to generate the final gaze behaviours. This would also allow us to improve our implementation of inhibition of return, which currently does not address how this effect is modulated depending on the intentions of the viewer.

Our methods are sensitive to the model parameters, and to the parameters of PSM which controls the saliency map generation. Parameter sensity and tuning for PSM was described in [17, 18]. For the particle model, it is important to choose appropriate sampling points, and gradient step size for the best results. Large step sizes result in targets being missed which can produce oscillations. The saliency decay rate, fixation duration, and fixation conditions should be chosen appropriately for the desired behaviour in both models. For example, larger decay rates and fixation durations and lower

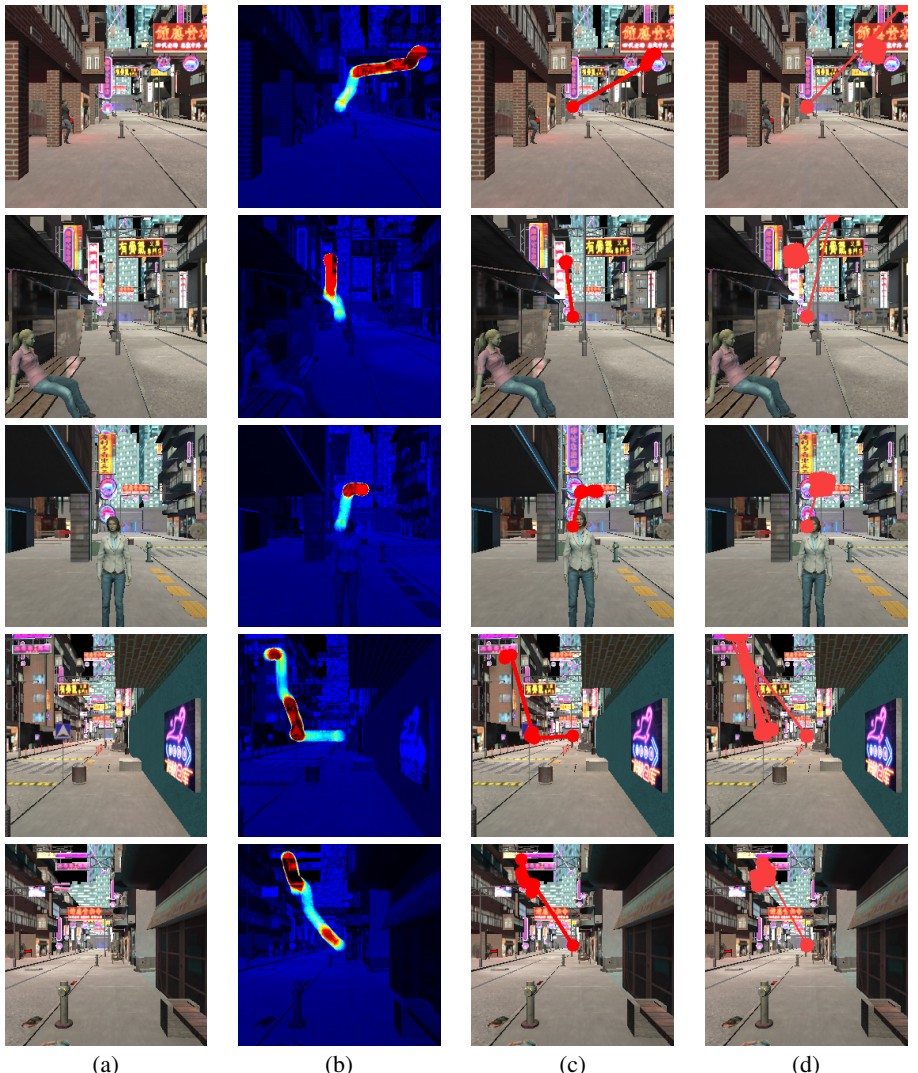

| (a) | (b) | (c) | (d) |

Figure 5: For (a) an RGB image, (b) gaze heatmaps for our particle method overlayed on the RGB image, comparison of scanpath traces between (c) our method with (d) pyStar-FC.

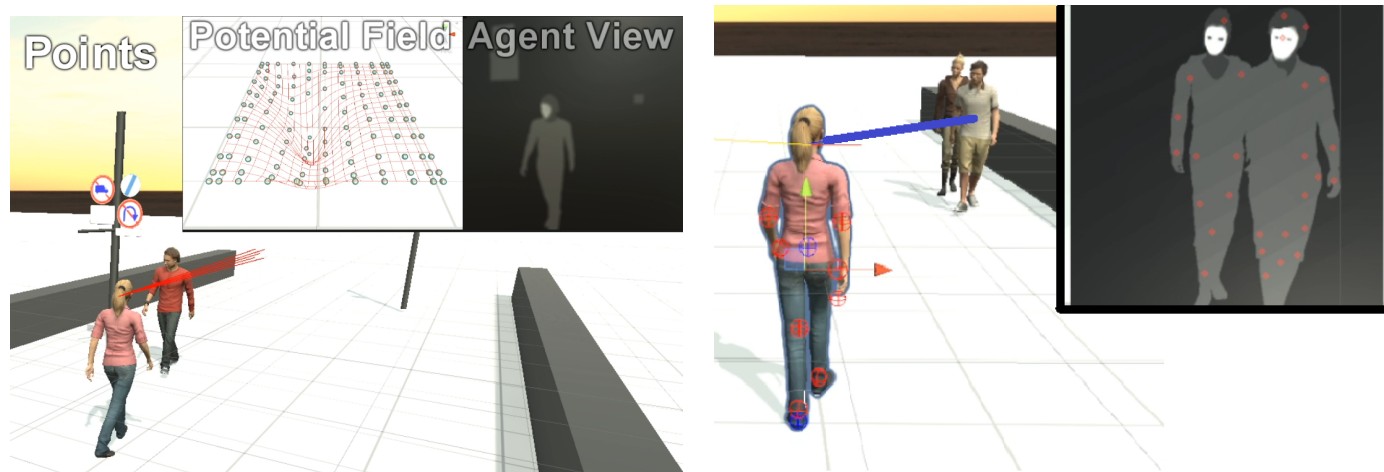

Figure 6: Comparison of target selection between the two models. Left: Particle model, which drives gaze in the direction of the potential field gradient. Right: Probabilistic model target selection. Red circles indicate potential target locations, which once selected will trigger a saccade.

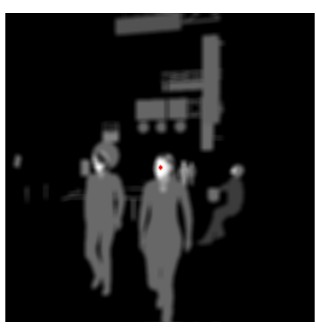 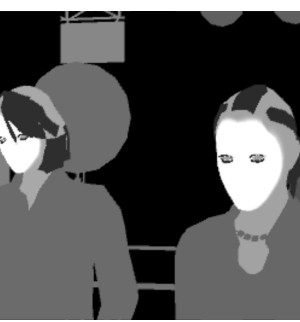

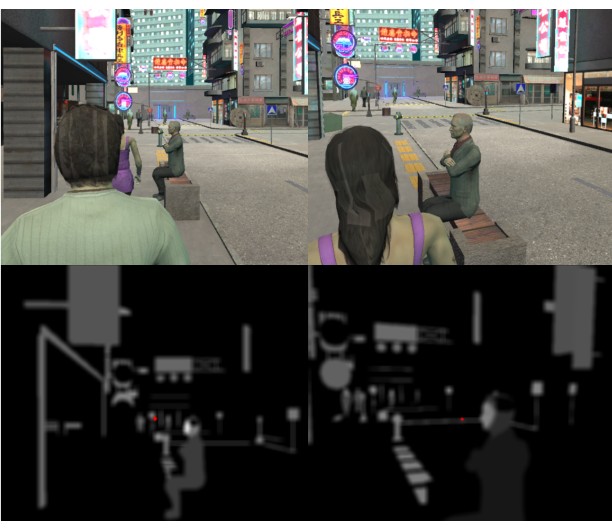

Figure 7: Left: Particle model saliency map. Right: Probabilistic model view of the same subjects. The Probabilistic model uses a reduced field of view to produce a zoomed effect for the purpose of facilitating microsaccades.

Figure 9: One agent walks behind another and both see the same man sitting on a bench. On the right the man's saliency is lower due to saliency decay during fixation. Top: RGB view from behind the agent so that the head orientation is visible. Bottom: Saliency map from the agent's POV using the particle model.

thresholds for triggering fixations produce quicker, darting gaze behaviours.

A limitation in both of our methods is that only the current view of the agent is considered, and objects outside the agent's current field of view do not impact gaze behaviour. The saliency decay mechanism models some aspects of memory, since the saliency amount is remembered even if it leaves an agent's field of view and then comes back into it. However, complete models would include a model of memory that keeps track of objects recently seen but not currently within the field of view and their relative positions so that agents could look back at them directly even when they are not inside the field of view. Matching our model parameters to real human gaze data and comparing it against other models remains important planned future work. However we have illustrated that our model is highly flexible and customizable, and can be used to author a variety of virtual gaze behaviours.

## 7 CONCLUSION

We presented two Saliency-Driven Gaze Control (SDGC) methods, the *particle model* and *probabilistic model*, which when combined with appropriately defined saliency (attention) are able to cover a wide range of well studied and understood human gaze-movements. SDGC takes as input a real-time map off attention in an autonomous agents visual field and generates gaze-movements. The two SDGC methods, the particle model and probabilistic model, are able to elicit physiologically based head movements, smooth pursuits, saccades and microsaccades. For a defined visual task, we show that through combination of parameterized visual attention and gaze-movements that appropriate gaze behaviours will arise.

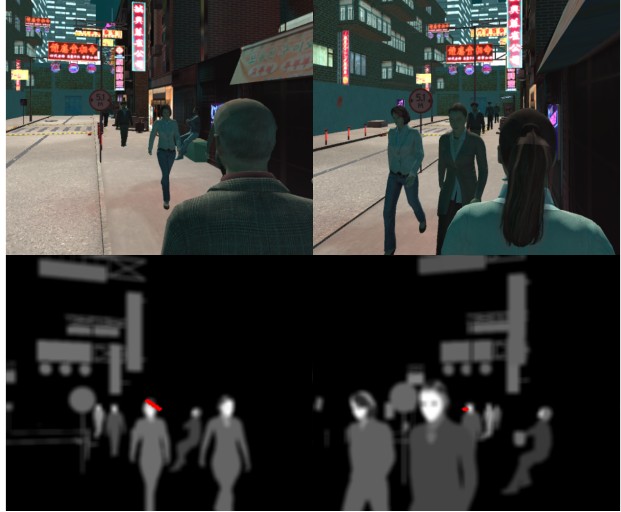

Figure 8: Two agents walking while using the particle gaze model simultaneously. Top: RGB view from behind the agent. Bottom: Saliency map from the agent's POV. The small red line in the center of the saliency map indicates the current direction of the particle gradient.

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
