# OpenReview forum: "Saliency Driven Gaze Control for Autonomous Pedestrians"
_graphicsinterface.org/Graphics_Interface/2023/Conference_SD — GI 2023 - second deadline_

### Official Review · Reviewer_VhB1 · 2023-04-21
**Presents interesting techniques for saliency driven gaze control, still needs some refinement and clarification**

**Rating:** 6
**Confidence:** 4

**Review:**

This paper presents techniques controlling the gaze of virtual avatars that integrate saliency maps generated by a parametric model.  Two control methods are explored. The first is based on a particle method that interprets the saliency map as a potential field  and a particle is integrated through the field using gradient information to gaze at minimum in the field, i.e. the most salient features. The second method interprets the intensity in the saliency map as a probability, and changes gaze targets based on a random sampling technique. There are smooth transitions between gaze targets, and a decay mechanism is used to cause target switching after some time.

I reviewed a previous version of this paper submitted for the earlier deadline. It is appreciated that the authors took the time to respond to reviewer concerns and questions in the current revised version. I am also encouraged by the authors' willingness to revise the manuscript to clearly convey the objectives and technical details of the their work.

New experiments were added that compare the behavior of the particle model to the pyStar-FC model. This not quite an "apples-to-apples" comparison since the pyStar-FC model is used to predict human gaze for real static images, whereas the proposed technique focuses on authoring gaze transitions from saliency maps. Rather I interpret this result as demonstrating how the proposed method can be integrated with existing models to generate gaze targets.  However, details behind such an integration are not discussed, and it would have been valuable to know details behind how the particle model can be tuned to match the output of pyStar-FC. The manuscript should be updated to reflect these steps.

Some justification is provide at the end of Section 4.2.1 as to why the saliency map is lifted to a spline surface, since this allows interpretation as a physics type problem. I am doubtful about this reasoning. What is really needed here is an ability to find gaze targets at minima in the potential field. I suspect a robust optimization technique, e.g. particle swarms, may perform just as well or perhaps even better in cases where the potential field is multimodal or avoid issues due to coarsely sampled spline control points, e.g. the agent gazes just off-to-the side of highly salient points as can be seen in the video.

It is unfortunate that Equation 1 was not revised with a (brief) explanation of terms in the equation.

I am still on-the-fence about accepting this paper, but leaning positive.

---

### Official Review · Reviewer_PCWz · 2023-04-25
**The evaluation has been enhanced, yet remains inadequate, and the work's technical significance is relatively limited.**

**Rating:** 4
**Confidence:** 2

**Review:**

In this paper, the authors present a gaze control framework based on a saliency map and introduce two distinct models. The first model converts a generated saliency map into a potential field, generating forces onto a simulated particle that acts as a proxy for gaze direction. The second model treats the saliency map as a probability distribution and interpolates the current gaze direction to a random point sampled from it. The authors claim that the framework is flexible and can be tuned to reproduce the performance of other frameworks.

This paper is a resubmission, and the new revision includes an additional comparison with a previous system. However, the justification provided still appears to be insufficient. The comparison with the baseline model focuses on the similarity of the fixation points computed by both models but lacks quantitative evaluation or user studies addressing the quality of the generated motion. Additionally, the discussion of the experiments is inadequate. The authors acknowledge that the models are sensitive to parameters; thus, it is crucial to provide more information about how these parameters are tuned to match the baseline. Is this tuning based on the "ten pairs of images" mentioned in the third paragraph of section 5? How does the tuned model perform on new images?

Furthermore, the models presented in the paper seem rather simplistic. The advantages of using the particle model in terms of gaze quality, as compared to a simpler approach like interpolating among saliency points, remain unclear. The work could gain significance if it demonstrates that the generated gaze motion is human-like quantitatively, or if it proposes a method for automatically tuning the parameters to match human gaze patterns. As it stands, the paper resembles a technical report and does not appear ready for publication.

---

### Official Review · Reviewer_BRy1 · 2023-04-25
**Another experiment added but I am not sure if that is strengthening the argument of this paper.**

**Rating:** 5
**Confidence:** 3

**Review:**


This is the second cycle of the review.   I mentioned about the poor presentation of the paper in the previous round.   The authors have added some comparison with another gaze model " pyStar-FC" in this version.

I am rather confused about the comparison with pyStar-FC that the authors are adding:  I thought the point of the comparison is to show the proposed method produces better results in terms of human gaze or head motion - on the contrary, the authors simply mention that the proposed method can produce results similar to  pyStar-FC.  Then why not just use  pyStar-FC from the beginning?

The authors should evaluate the motion of the head or animation that shows the improved realism of the animation due to the usage of their gaze model.  There is no evaluation or qualitative test like that in the paper.   The video shows some head motion of the characters watching different directions, but this is rather subtle, and it is difficult to judge if the resolution of the head area is low.

The authors mention about the potential interesting future work - these are interesting, but isn't there any evidence that this model is most suitable for such applications compared to other models?


"  page 6: First, it should be noted
that the PSM saliency maps our models are predicated on have been
previously evaluated against SALICON,

page 6: so we construction scenarios in our virtual
environment

page 6: ior (inhibition of return)　-> IOR

page 7:  Matching it to real human gaze data should therefore be possible and is important planned future work.
-> this claim sounds very optimistic and without any justification